# The Brain Research Hotspot Database (BRHD): A Panoramic Database of the Latest Hotspots in Brain Research

**DOI:** 10.3390/brainsci12050638

**Published:** 2022-05-12

**Authors:** Pin Chen, Xue Lin, Anna Liu, Jian Li

**Affiliations:** 1The Key Laboratory of Developmental Genes and Human Disease, Ministry of Education, School of Life Sciences and Technology, Southeast University, Nanjing 210018, China; 220193518@seu.edu.cn (P.C.); 220193506@seu.edu.cn (A.L.); 2Department of Bioinformatics, School of Biomedical Engineering and Informatics, Nanjing Medical University, Nanjing 211166, China; xue.lin@njmu.edu.cn

**Keywords:** brain science, database, disease-gene network, brain and gut microbiome, 3D brain structure study

## Abstract

Brain science, an emerging, dynamic, multidisciplinary basic research field, is generating numerous valuable data. However, there are still several obstacles for the utilization of these data, such as data fragmentation, heterogeneity, availability, and annotation divergence. Thus, to overcome these obstacles and construct an online community, we developed a panoramic database named Brain Research Hotspot Database (BRHD). As of 30 January 2022, the database had been integrated with standardized vocabularies from various resources, including 423,681 papers, 46,344 patents, 9585 transcriptomic datasets, 261 cell markers, as well as with information regarding brain initiatives that were officially launched and well-known scholars in brain research. Based on the keywords entered by users and the search options they set, data can be accessed and retrieved through exact and fuzzy search scenarios. In addition, for brain diseases, we developed three featured functions based on deep data mining: (1) a brain disease–genome network, which collects the associations between common brain diseases, genes, and mutations reported in the literature; (2) brain and gut microbiome associations, based on the literature related to this topic, with added annotations for reference; (3) 3D brain structure, containing a high-precision brain anatomy model with visual links to quickly connect to an organ-on-a-chip database. In short, the BRHD integrates data from a variety of brain science resources to provide a friendly user interface and freely accessible viewing and downloading environment. Furthermore, the original functions developed based on these data provide references and insights for brain research.

## 1. Introduction

Brain science is a vigorously developing field in basic research. It has a leading role in improving the treatment of brain diseases, protecting brain health, and elucidating neural mechanisms at various levels [1]. In recent years, with the official launch of a series of large-scale brain research projects led by countries or organizations, brain research has achieved fruitful results.

The Allen Institute for Brain Science has a leading position in the realization of the brain atlas and in cell science. In 2016, the Allen Institute for Brain Science presented a map of the human brain structure that was the most accurate at the time, based on the brain of a 34-year-old healthy woman [2]. In 2021, the researchers from Allen Institute for Brain Science successfully applied expansion sequencing (ExSeq) to a map of the mouse brain, improving its spatial resolution from nanoscale to system level [3]; this resolution has the potential to reveal the spatial pattern of gene expression in the context of cancer biology and immunology. A gratifying breakthrough in neurodegenerative disease treatment was achieved, for example, the application of positron emission tomography and the development of easy-access and low-cost blood-based biomarkers for the detection and diagnosis of Alzheimer’s disease [4]. Breakthrough progress has also been made in the research of brain-inspired intelligence. In 2019, the world’s first heterogeneous fusion brain-inspired computing chip “Tianjic” was developed by Tsinghua University and was mounted on autonomous bicycles for display [5].

On the whole, brain science is on the eve of a revolutionary breakthrough. Brain science research is now moving from the macroscopic understanding of the functions of various brain regions to the mesoscopic clarification of the brain structure, and then to the analysis of neuron structure and function at the microscopic level [6]. With the development of brain science, a large amount of valuable data were generated, which led to the creation of novel brain science databases. These include those of The UCL Dementia Research Centre [7], The Alzheimer’s Disease Neuroimaging Initiative (ADNI) [8], The National Institute of Mental Health Data Archive (NDA) [9], which contain data on brain diseases, as well as those of The Allen Institute for Brain Science [10], BrainMap [11], Neurosynth [12], OpenNeuro [13], The Open MEG Archive (OMEGA) [14], which provide brain imaging data, and those of the NeuroImaging Tools & Resources Collaboratory (NITRC) [15], which provides brain analysis tools, and of The Neuroscience Information Framework (NIF) [16] and The Connectome Coordination Facility (CCF) [17], which provide neural pathway data. These databases play a significant role in supporting the development of brain science.

However, there are still several obstacles in the utilization of brain science data: their fragmentation, heterogeneity, complexity. Data fragmentation means that different types of brain science data are scattered in various databases. A comprehensive brain research database is needed, which can greatly reduce the time spent by researchers in the data collection process, thus enabling researchers to obtain a preliminary and comprehensive understanding of current brain science research before conducting experiments. Regarding data heterogeneity, due to the fragmentation of the current data, different types of annotations leading to different formats are difficult to combine. As for data complexity, the current brain science data are cumbersome and messy and lack visual information, so it is difficult for users to read them. To construct an online community that provides resources without these hurdles, we constructed the Brain Research Hotspot Database (BRHD, accessed on 2 May 2022, http://www.brainresearch.cn/), a comprehensive database focusing mainly on brain diseases.

## 2. Results

### 2.1. A Panoramic Database of Brain Science

In this study, we aimed to construct a panoramic database of the latest research hotspots in brain science. To achieve this aim, we used both manual and automation tools to search and retrieve brain research data from public databases. As of 30 January 2022, we collected 423,681 papers, 46,344 patents, 9585 transcriptomic datasets, 261 brain cell markers [18] using Boolean logic search of the information provided by officially launched brain initiatives [19,20] and well-known scholars in brain research. We performed natural language processing (NLP), manual annotation, standardization, and visualization of these data. The processed data constituted the blueprint of the BRHD (as shown in Figure 1). Brain development disorders, mental diseases, and neurodegenerative diseases are increasingly threatening human health [21]. According to the latest statistics of the World Health Organization (WHO) on brain health, neurological and neurodevelopmental diseases are a heavy medical burden worldwide and cause 9 million deaths each year; this makes them the second leading cause of death globally [22]. In addition, for brain diseases, we developed three featured functions based on deep data mining: a disease–genome network, Brain and Gut Microbiome which contains the brain and gut microbiome associations, 3D brain structure study. Thus, this webserver will provide users with featured resources and popular resources.

### 2.2. Featured Resources

The Featured Resources are the most significant elements of the BRHD. After unique annotation and visualization, we provide references and insights for precision medicine, brain disease diagnosis and treatment, brain organ chip development.

#### 2.2.1. Disease–Genome Network

Genetic information regarding human diseases is decisive in precision medicine and drug discovery [23]. A new gene therapy method involving MRI-guided convection-enhanced delivery (iMRI-CED) will greatly improve the treatment of neurodegenerative diseases [24]; the key step of drug discovery is the identification of new targets or biomarkers, which requires the support of genetic information on the disease of interest [25]. Therefore, we identified and listed the genes and mutations related to the brain diseases, then found links between each disease and genes or mutations reported in papers and in professional representative databases, i.e., Online Mendelian Inheritance in Man (OMIM) [26], Ensembl [27], MalaCards [28]. Our database reports the associations between 34 brain diseases, 1156 genes, 7065 mutations, then creates a brain disease-associated genome network map to explore human brain diseases and corresponding genes, mutations, and interrelationships at the higher levels of cells and organisms [29], as shown in Figure 2. Users can select three search types in the search box, i.e., disease, gene, variant. Two types of search results will appear according to the three search types, which will be disease-associated genes and disease associated-variants, respectively. Clicking the hyperlink of the returned results will redirect the users to a detailed web page, which will provide information on Brain disease, Associated gene, Gene full name, Gene category, Associated variant, Supporting evidence PMID, etc. The disease genome network section can generate dynamic visualization charts based on the user’s search keywords. Figure 3 shows in the middle the retrieved brain disease, while the dots next to it represent related genes. The larger the gene dot, the larger the literature supporting the relationship between that gene and the disease. Figure 4 shows the gene network map based on data reported in the literature that we created and appears to be related to two or more brain diseases (this map does not indicate that the genes reported in the literature are related to a single brain disease).

#### 2.2.2. Brain and Gut Microbiome

Research shows that there is a definite “brain–gut axis” between the gut microbe and the brain, but its regulation and mechanisms of function are still unclear [30]. Some studies have shown that the intestinal microbes play a significant role in the formation of the blood–brain barrier, as well as in myelination, neurogenesis and behavioral regulation and other basic neural processes [31]. Moreover, germ-free mice that lack any microorganism will behave differently from traditional mice in various aspects, as they were shown to take greater risks and display hyperactivity and defects in learning and memory [32,33,34,35]. We investigated the association between the brain and the gut microbiome and collected references and insights from studies of the brain–gut axis. We also made unique annotations to the collected 198 papers manually, regarding the upregulation or downregulation of microbes, as well as some key microbial information reported in clinical trials related to brain diseases (as shown in Figure 5). The information we provide in our database includes two sections: literature information and experimental information. The literature information includes the items PMID, Title name, Journal name, Publication year, PubMed Link, whereas the experimental information includes the items Microbiome, Associated brain region, Associated phenotype, Experimental condition, Sample number, Qualitative outcome, Experiment method, Location, Conclusion, Experiment platform, Statistical method. As of 30 January 2022, we had manually collected 198 original research papers on the associations between brain and microorganisms from 3326 retrieved papers, sorted and then annotated them. Extensive information was extracted. Since most of the data provided in the literature are at the genus level, unifying this information to the genus level is also clinically instructive; therefore, we normalized the expression levels of these microorganisms to the genus level [36].

#### 2.2.3. Resource for the Study of the 3D Brain Structure

Organ chips are an important part of brain-inspired technology and provide better models for disease modeling, organ transplantation, and drug screening [37]. In the future, it is expected that it will be possible to reconstruct the brain in vitro using the organ chip technology. Organ chips have many applications in brain science. For instance, a cortical organoid was used to test the effects of the anti-epileptic drug valproic acid (VPA) on the nervous system of offspring [38], a brain organoid to simulate the characteristics of Alzheimer’s disease (AD) and help drug screening [39], a cortical organoid to simulate the process of brain development [40,41], and a brain organoid to simulate human brain wrinkling related to neurodevelopmental disorders [42]. To provide such information and a simple and visible searching tool, we provide in our database a high-quality 3D brain anatomy model. Users can click on the brain regions of the model to be directed to the corresponding brain organ chip research data (as shown in Figure 6). The 3D brain anatomy model consists of 113 brain regions associated with the latest related database resources, including the literature on organ chips related to each specific brain region and a hot map of the literature, which can help the users understand the latest progress in the field of brain organ chips.

### 2.3. Popular Resources

The popular resources include a very large number and scope of basic resources for practical applications in the field of brain science. They comprise the databases of Research frontier, Brain initiative, Scientists, Transcriptome analysis, Patent, Cell marker.

#### 2.3.1. Research Frontier

The Research frontier database contains 423,681 publications on brain science. We searched in the Web of Science Core Collection [43] records of the last 10 years, including basic information such as title, author, abstract, publishing information, and the links to the full texts in the original sources. Then, the related public data were retrieved from open-access databases like PubMed, etc. In the BRHD, the data from Research Frontier can be searched through ordinary search and advanced search. Using ordinary search, publications can be retrieved by keywords, also using multiple keywords and their combinations. In addition, information, such as type of the paper and year of publication, is also included and can be used for searching. The results will present more detailed information including article title, authors’ full names, source title, volume, issue, DOI, document type, full text link, other information.

#### 2.3.2. Brain Initiative

The Brain initiative database contains information on brain science projects that have been put forward and implemented by major countries, including brain projects from six countries or regions, i.e., the United States, the European Union, Japan, Australia, Canada, and South Korea, as well as seven well-known brain science projects, such as the blue brain project. It mainly includes the details of each brain science project, information regarding the leading scientists, the latest progress, the timeline of the plan. It records each subplan of these brain science projects and the key time points for subsequent developments and improvements.

#### 2.3.3. Scientist

In this section, the list of scholars is based on 10,122 highly cited papers in brain science in the recent 10 years (highly cited papers refer to papers that are in the top 1% of the works most cited in the world in the past 10 years). CiteSpace 5.8.R3 was used for co-citation analysis and literature co-publication analysis. We collected and filtered out the relevant information, mainly presenting information such as institution, education, publication(s), contact of the main involved scientists.

#### 2.3.4. Transcriptome Analysis

The database includes 3416 transcriptomic datasets from the Array Express database [44] and 6169 transcriptomic datasets from the Gene Expression Omnibus (GEO) [45] database regarding the last 10 years, helping users to access brain transcriptomic data. In this web page, the users can select the Array Express or GEO database, as well as experiment type and species, i.e., Mus musculus, Homo sapiens, Rattus norvegicus, to filter the data. The users can also search with keywords or their combinations through the search bar. Title, organization, experiment type, summary, platforms and other platform data, as well as data of soft formatted family file(s), minimal formatted family file(s), series matrix file(s) will be returned as the searching result.

#### 2.3.5. Patent

The Patent database contains 46,344 invention patents related to brain science with patent details and the hyperlinks to the corresponding full texts. The patent numbers were obtained from the Derwent Innovations Index [46] using system keywords, then were used to query and sort the patent details from Free Patents Online [47]. In the Patent resource, users can filter the results on the basis of the patent type and search for patents by keywords. We also provide hyperlinks to patents’ titles, abstracts, assignments, claims, document types and numbers, international classes, and original texts.

#### 2.3.6. Cell Marker

The Cell Marker contains information of 124 human brain cell markers and 137 mouse brain cell markers [18]. Users can select the animal species, tissue type, cell type, and then the system will display the relevant cell markers and the literature information supporting the record. Through this function, users can quickly find the brain cell markers of interest and the related experimental and literature information.

## 3. Methods

### 3.1. Overview of the BRHD

The workflow for the construction of the BRHD database is presented in Figure 1. The first step was to summarize the key words used in brain science. As the research of brain science covers a wide range of scientific studies, there is no ready taxonomy for brain research. We thus summarized the keywords of brain research representing various aspects of the development of brain science from different reliable resources, i.e., Wikipedia, Medical Subject Headings (MeSH), and related literature with the goal to collect enough representative and comprehensive information. The second step was to use these representative keywords and automation tools to obtain the required resources from public databases and manually obtain data from various official websites. The third step was to process the obtained data. We performed Extract–Transform–Load System processing on the collected raw data, organized them into specified formats, and unified these formats after processing, referring to authoritative databases such as NCBI Taxonomy for calibration. The fourth step was to transmit the collected data through file transfer tools, such as Flume-FTP and Apache Kafka, to the data interface. We extracted documents, patents, experts’ names, transcriptome information, genetic information, microbial information in the texts. The processed data were stored in distributed storage databases through MySQL we finally obtained 6 basic databases and 3 featured databases.

### 3.2. Data Acquisition

We used summarized systematic keywords to collect brain science data. The network of the systematic keywords is shown in Figure 7. The BRHD integrates data from expert curated databases with information collected from text mining scientific literature, hoping to provide users with a comprehensive landscape of brain science research.

The BRHD includes data from the following resources: PubMed [48], Web of Science Core Collection [43], Derwent Innovations Index [46], Free Patents Online (FPO) [47], Array Express [44], Gene Expression Omnibus (GEO) [45], Cell Marker [18], Online Mendelian Inheritance in Man (OMIM) [26], Ensembl [27], MalaCards [28], etc. Take the brain and gut microbiome for example, we used “Brain” and “Gut” and “Microbiome” as keywords to search in PubMed and we retrieved 3326 publications. After a manual check, 198 publications were classified as related, then a fields table was prepared, designed to extract important information from the 198 publications. Two types of data were collected, i.e., changes in the expression levels of the gut microbial community in individuals with brain diseases and experimental information regarding the reported cases with brain diseases that were treated with gut microbes. Finally, the extracted information was standardized by comparison with popular controlled vocabulary thesaurus and taxonomy databases, such as NCBI Medical Subject Headings (MeSH) [49] and NCBI Taxonomy [50].

### 3.3. Intellectual Property of the Involved Data Sources

The data sources of our database are mainly major open-source research databases. Data were downloaded and mined according to the regulations of the original databases, and the sources and the original website links were uniformly marked in a prominent place. The non-local content of our database only shows relevant information such as abstracts, not the full text, and provides information on original sources and links.

Research frontier: We used the summarized keyword groups to conduct Boolean logic searches in the Web of science core collection (accessed on 1 December 2021, https://www.webofscience.com/wos/alldb/basic-search), retrieved PubMed ID information and accompanying relevant information (accessed on 1 December 2021, https://pubmed.ncbi.nlm.nih.gov/) using the PubMed ID and organized it.

Scientist: Based on the results of the literature search, we used CiteSpace to analyze and obtain open-access information on the most popular scientists in brain science research, then the information was used to manually obtain open-access information of the general publications on the official websites for various scientists and the information was organized.

Brain initiative: We summarized the information of large-scale brain science projects confirmed to be launched around the world from literature, then the related open-access information was obtained manually from the official websites of these brain projects with citations of the original sources.

Transcriptome analysis: We used the summarized keyword groups to obtain various transcriptome data from Gene Expression Omnibus (GEO, accessed on 15 December 2021, https://www.ncbi.nlm.nih.gov/geo/) and ArrayExpress (accessed on 15 December 2021, https://www.ebi.ac.uk/arrayexpress/) and organized them.

Patent: We use the summarized keyword groups to retrieve Patent Numbers from the Derwent Innovations Index database (accessed on 26 December 2021, https://www.webofscience.com/wos/alldb/basic-search), then used the obtained Patent Numbers information to collect detailed information of each patent from Free Patent Online (accessed on 26 December 2021, https://www.freepatentsonline.com/). The original sources were thoroughly cited.

Cell Marker: Data from this section were mainly from the CellMarker database (accessed on 16 November 2021, http://biocc.hrbmu.edu.cn/CellMarker/index.jsp) and were combined with data summarized from the literature. The original sources were thoroughly cited.

Disease–genome network: We used the summarized keyword groups to manually obtain the associations and supporting data between diseases and genes and mutations from the Online Mendelian Inheritance in Man database (OMIM, accessed 25 November 2021, https://omim.org/), MalaCards database (accessed on 26 November 2021 https://www.malacards.org/), then aggregated them and used Cytoscape software to visualize and analyze them. The original sources were thoroughly cited.

Brain and gut microbiome: We used the summarized keyword groups to search and download original research literature on brain and gut microbiome on PubMed (accessed on 30 November 2021, https://pubmed.ncbi.nlm.nih.gov/), then manually reviewed and annotated the formation. The original sources were thoroughly cited.

3D brain structure study: We set up a high-precision 3D brain anatomy model and linked it to the cooperative Organ on a chip Database (accessed on 1 December 2021, http://www.organchip.cn/) to integrate the latest research on brain anatomical sections related to organ chips.

### 3.4. Data Processing and Management

The collected raw data were processed using the Extract–Transform–Load System and natural language processing. Brain science information and relational models were extracted from the processed data and stored in distributed storage and search engines, i.e., MySQL 8.0.15 and Elasticsearch 6.4.0.

The main tables in our database include: the ‘Brain Research Hotspot Database Dictionary’ table, popular resource tables (i.e., ‘Research Frontier’ table, ‘Scientist’ table, ‘Brain Initiative’ table, ‘Patent’ table, ‘Transcriptome analysis’ table, and ‘Cell Marker’ table), featured resource tables (i.e., ‘Brain disease-genome network’ table, ‘Brain and gut microbiome’ table, and ‘3D brain structure study’ table). These tables are related to each other through multiple attributes such as ‘Brain Initiative Name’, ‘Scientist ID’, ‘Patent ID’, ‘PubMed ID’, ‘Cell Marker ID’, ‘Transcriptome data ID’, ‘Brain anatomy’, ‘Disease’. The structure and detailed relationship of the tables are presented in Figure 8. By this means, the literature information in ‘Research Frontier’ provides evidence for ‘Cell Marker’ to determine whether a certain small molecule is a marker of a certain cell. The brain research hotspot scientists in ‘Scientist’ are summed up by using CiteSpace software to analyze the literature of Research Frontier. The literature in ‘Research Frontier’ plays a role as a data support. Each part of the brain anatomy structure on the virtual model in the 3D brain structure study resource is linked to the related literature information in the ‘Research Frontier’ table, which provides evidence to understand if a certain microbe is related to the brain and thus offers information related to the ‘brain and gut microbiome’.

For the collection of brain science data, we adopted a series of standardized procedures to ensure the comprehensiveness and accuracy of the data. First of all, the data we collected were from various major brain research fields, with a focus on brain diseases. Due to the wide variety of brain diseases, classification standards are different. In order to establish a scientific and comprehensive literature collection method, we summarized systematic brain research keywords with brain diseases as the main content from papers [51,52,53], Medical Subject Headings (MeSH) [54], and Wikipedia and the corresponding various English annotations, then mapped the systematic brain research keywords network, as shown in Figure 7. Then, we used these keywords to retrieve and collect the corresponding data from multiple databases, mainly using a combination of intelligent programs and manual screening.

We employed strategies for data curation and verification. Take annotation as an example, we mainly annotated the brain and microbiome data. The original research articles were retrieved using the keywords ‘Brain’, ‘Gut’, and ‘Microbiome’ from PubMed. The retrieved articles were then read through manually to summarize the appropriate annotations. Then, the summarized annotations were calibrated through standard databases, such as NCBI Taxonomy, to obtain standardized annotations. Each annotation was checked by two professional annotators. If there was disagreement between the two annotators, a short discussion of the entry was organized between the two annotators and the one who designed the annotation workflow and performed it. If an agreement was reached by at least two of the three involved scientists, the annotation was set. Otherwise, more information was obtained for further discussion. The annotation was abandoned if no agreement was reached after several rounds of discussion, which was relatively rare in our work.

Next, we performed Extract–Transform–Load System processing on the collected raw data and performed natural language processing on the processed data and sorted them into files, pictures, and other formats and then through Apache Flume 1.9.0, Core FTP LE 2.21, and other file transfer tools. We used Apache Kafka 2.13 and data interface transmission to collect the data. We selected information such as literature references, patents, experts, transcriptome information, genetic information, microorganisms’ information from the text, then built models of disease–genome and disease–microbiome relationships based on data types and their interrelationships, supported by biological theories. The processed data were stored in distributed storage systems through MySQL 8.0.15 and Elasticsearch 6.4.0. The collection of all these data was manually checked in detail by two independent researchers.

Apache Flume is a distributed mass log collection, aggregation and transmission software. Its function is to collect data from a data source (source) and then send the collected data to a specified destination (sink).

Core FTP LE is a file transfer protocol and one of the protocols in the TCP/IP protocol suite. This protocol is the basis for Internet file transfer. It consists of a series of specification documents. It enables users to log in to a remote computer through the FTP function, download the required files from other computer systems or upload their own files to the network.

Apache Kafka is an open-source distributed event-streaming platform. Event streaming is the practice of capturing data in real time form event sources like databases, sensors, mobile devices, cloud services, and software applications in the form of streams of events. It strings these event streams durably for later retrieval, manipulated, processed, and reacts to the event streams in real time as well as retrospectively, and routes the event streams to different destination technologies as needed.

### 3.5. Construction of the BRHD Webserver

The interface of the web server and the instructions for the functions of the web server are shown in Figure 9. The server runs JS and JQuery to capture the input and settings of the filter conditions the users set on the web page. A request is raised to be processed in Controller layer; after the process of the Service layer, the DAO layer will be called. Then, the database is accessed with MyBatis technology to request the data and return them to the web page for display.

## 4. Discussion and Conclusions

Brain science studies the neural basis of cognition, thinking, consciousness, and language, whose comprehension is the ultimate challenge for human beings to understand nature and themselves. Moreover, according to data from the World Health Organization (WHO), about 70% of the medical burden occurs in low-income and middle-income countries, while these countries’ responses to neurological diseases are far from adequate [22]. At the same time, the pathological mechanisms of most brain diseases are not completely clear, and there is a lack of effective indicators for early diagnoses, which makes the problem of brain diseases very serious. In recent years, thanks to the attention and investment of the international community in brain science, the development of high-throughput sequencing technology has enabled us to determine the root causes of some brain diseases faster and more cost-effectively [55]. The production of a large number of valuable data has brought improvements in all aspects of drug development, precision medicine, and brain disease diagnosis and treatment. Therefore, we built the Brain Research Hotspot Database (BRHD) to collect, analyze, organize, and manage these data, providing services publicly to global users.

The BRHD will be regularly updated every three to six months. There is a stable domain name and also maintenance personnel to keep collecting comprehensive brain science data, especially original data from experiments and data produced by our group to improve the database and provide users with a better experience. The maintenance program, i.e., index defragmentation, log file maintenance, file/data compaction, integrity check, will be run for the back-end relational database to keep the database and server healthy. The analysis group will move on to discover the potential connections between the data by adding more annotations, so to construct an ordered knowledge graph, while the back-end group will deploy more online tools to meet the diverse needs of users in future releases, in the hope that the BRHD can play a role in all aspects of brain research.

## Figures and Tables

**Figure 1 brainsci-12-00638-f001:**
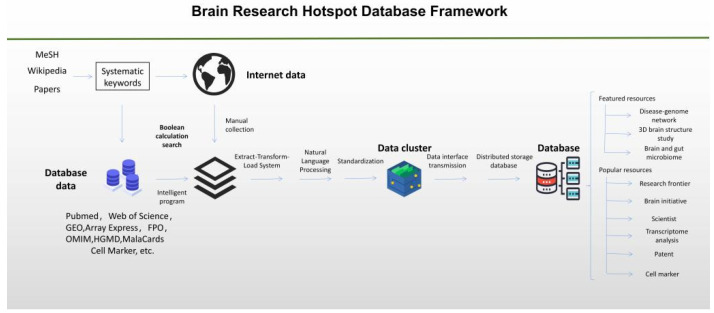
The framework of the BRHD. It mainly includes three parts: data collection, natural language processing, database creation.

**Figure 2 brainsci-12-00638-f002:**
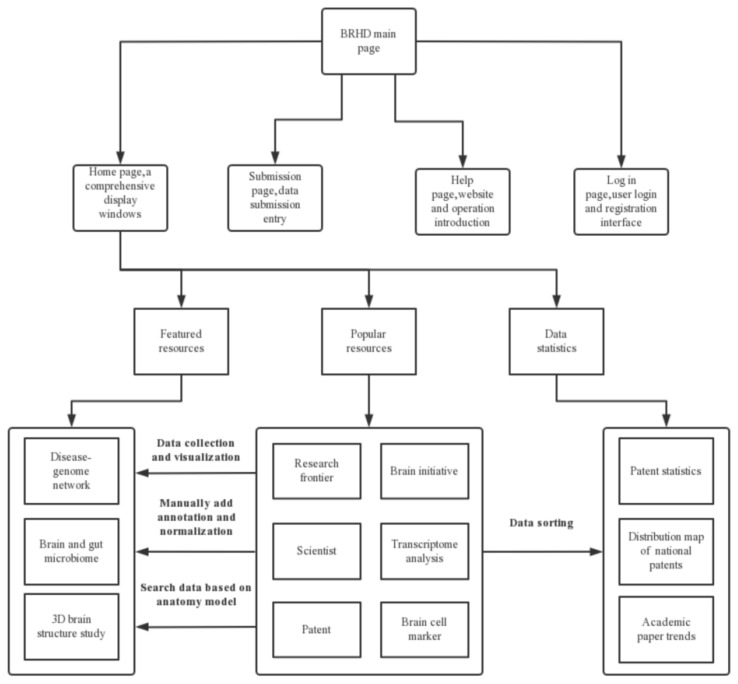
The main components of the BRHD. The BRHD is composed of popular resources and featured resources. Popular resources include the databases of Research frontier, Brain initiative, Scientists database, Transcriptome analysis, Patent, Cell marker. Featured resources includes Disease–genome networks, Brain and gut microbiome relationships, 3D brain structure. There are four tabs in the web server, which are Home page, Submission page, Login page, Help page.

**Figure 3 brainsci-12-00638-f003:**
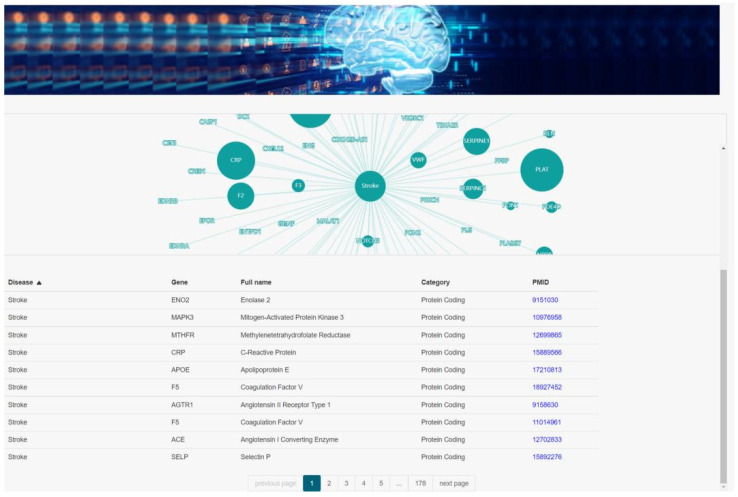
Page reporting detailed results of the retrieved genes related to a disease. The top visualization shows the link between disease and gene, with larger dots representing genes indicating more literature supporting the link. The lower part shows the specific annotation information.

**Figure 4 brainsci-12-00638-f004:**
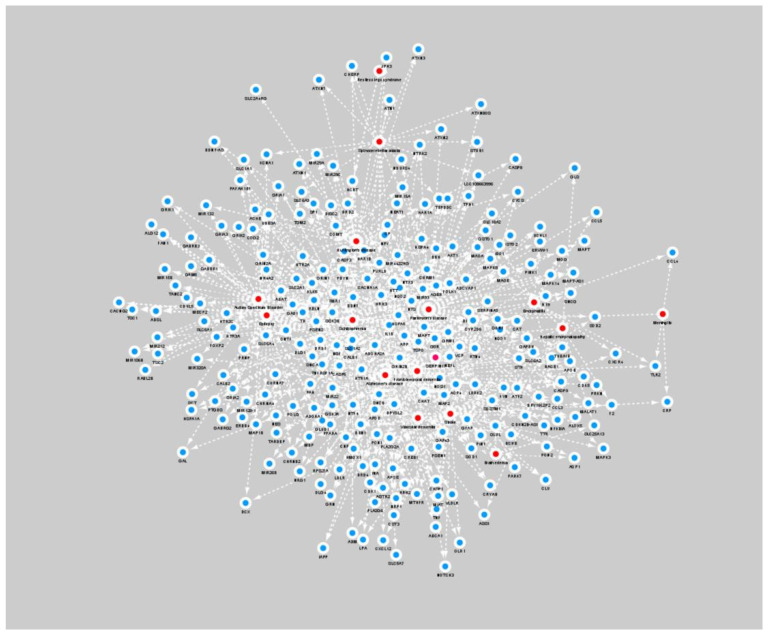
Network of disease–gene associations. Each red node in the figure represents a brain disease, each blue node represents a gene, and each line represents an association between a brain disease and a gene.

**Figure 5 brainsci-12-00638-f005:**
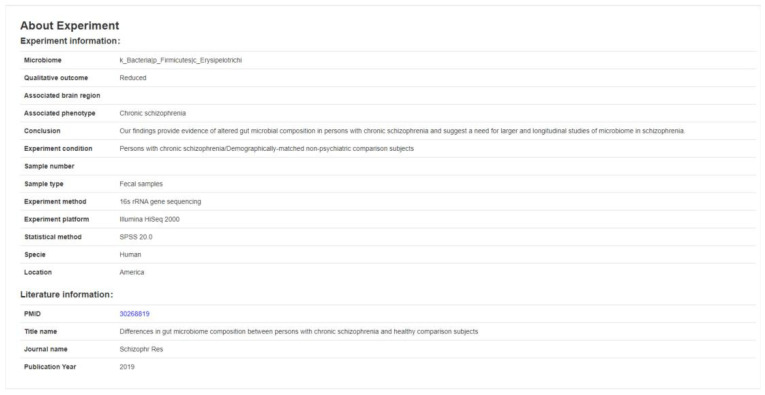
Display page of the content list regarding the brain and gut microbiome search interface. Users can search and read all the data collected in this section; experimental information and literature information of each record for reference is also displayed.

**Figure 6 brainsci-12-00638-f006:**
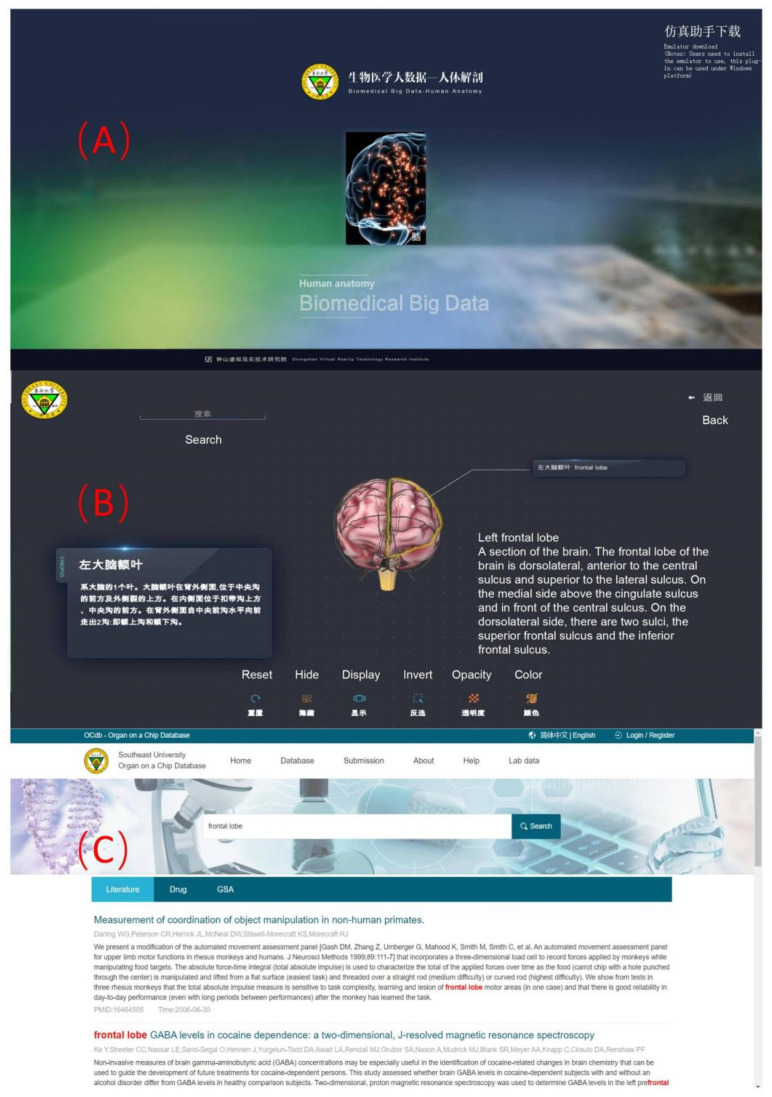
Web pages of the 3D brain structure study resource. (**A**). Welcome page of the 3D brain structure resource interface. (**B**). A high-quality 3D brain anatomy model. The user can rotate this 3D anatomical model to explore the structure of the brain or locate a part of the brain through retrieval, link to the related organ chip database, and obtain the latest organ chip development information. (**C**). Literature information on organ chip development related to the region of interest is provided in the database.

**Figure 7 brainsci-12-00638-f007:**
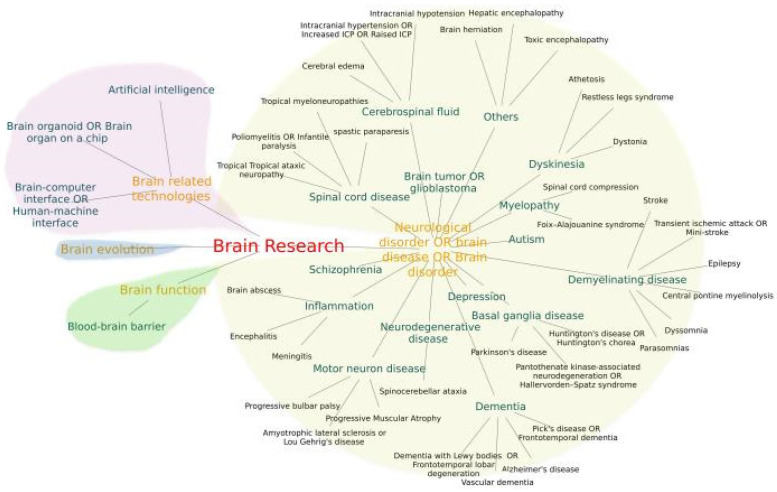
Systematic keywords network. Summarized keywords for systematic brain research. The different background colors of the branches of the networks presenting different fields of brain research.

**Figure 8 brainsci-12-00638-f008:**
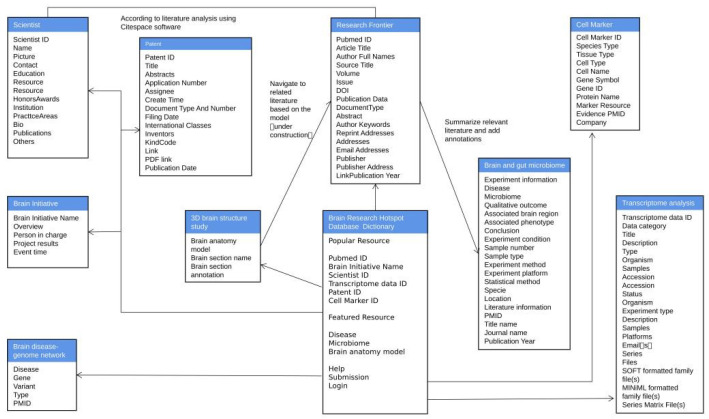
Structure of the relational database. The tables and relationships of the main datasets in the relational database of the BRHD.

**Figure 9 brainsci-12-00638-f009:**
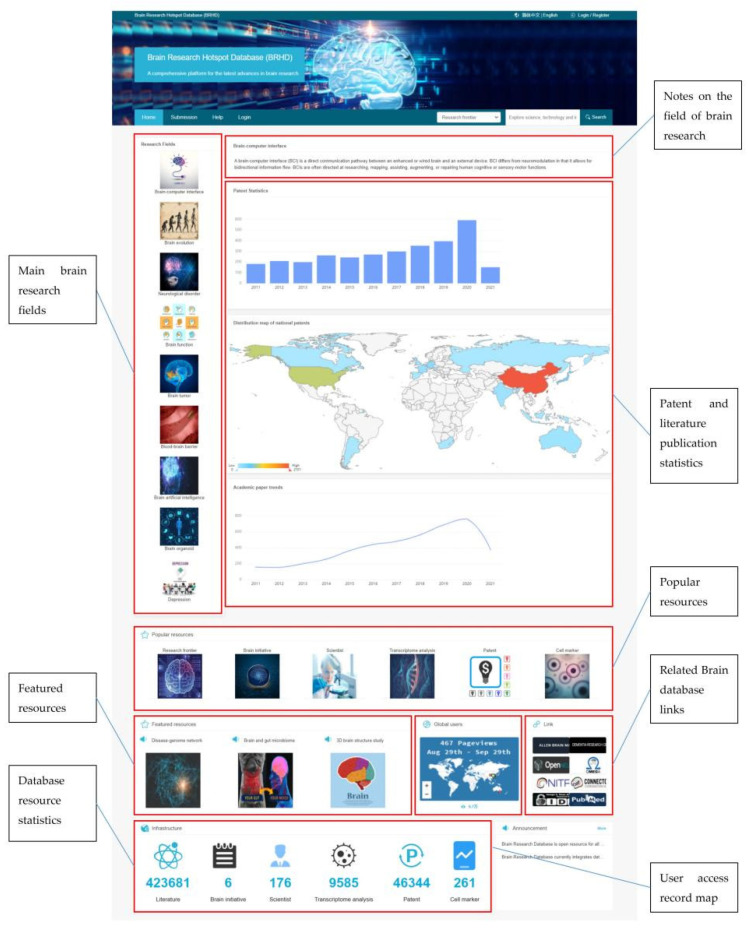
Introduction of each part of the BRHD main page.

## Data Availability

The data presented in this study are available in the webserver BRHD, which is freely accessible (accessed on 2 May 2022, http://www.brainresearch.cn/).

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
