# Peer review of "The Brain Research Hotspot Database (BRHD): A Panoramic Database of the Latest Hotspots in Brain Research"

_brainsci, 2022, doi:10.3390/brainsci12050638_

Round 1

Reviewer 1 Report

This revision addresses all my concerns.

Author Response

Response to Reviewer 1 Comments

Reviewer 2 Report

The authors report the design and implementation of the BRHD. In general, I found the effort rather interesting and applaud the authors for their effort. At the same time, however, I am left with a feeling that the primary contribution is one of harvesting the efforts of others. Although such 'accumulation' efforts are often valuable, I am not entirely convinced that care was taken to ensure all intended nuance in the original efforts was allowed to flow through the BRHD filters. Furthermore, saving some software and database development effort, I am not convinced that the authors added a substantial degree of novelty here.

Author Response

Response to Reviewer 2 Comments

Reviewer 3 Report

The authors present BRHD, a panoramic database of latest research hotspots for brain. It provides a comprehensive resource of information regarding "brain science". A lot of automatic and manual efforts have been done to build such resource. Although the database doesn't report new data, this is a useful resource as it collects many information gathers from the major public databases and literature, in a single portal.
One of the major issue of this kind of database/portal is the maintanance and sustanaibility in mid-log time especially if the updates require many human intervention and personnel. However the authors, in the conclusion, declare to have the necessary efforts for its sustanaibility.

Minor Comments
-Disease-genome network (line 127-130)
It seems, navigating the portal, that the BRHD doesn't visualized a network of connections created by the query (via Cytoscape), but it visualizes the same static figure of the same network indipendently from the query results. Please explain this behaviour and modify the text in the publication accordingly.

Line 273
The phrase "After adding a crawler task, first start to fill in the basic information of the crawler. After filling in the basic information, start the configuration" is not clear, please rewrite it. What are the configuration and the basic information? 

Data processing and Management (Line 343-343)
My opinion is: in general, the relational models are build from the data and their relations and not extracted, as written in the sentence

Line 388-389
too many "and" in the sentence. It is not clear the role of Apache flume, Core FTP, Apache kafka. Please rewrite the sentence clarifyin their role and use

Line 393
what are the "other databases", so far only MySQL DBMS has been indicated.

Figure 2 caption: line 133 "Research Frontier" instead of "Frontier Research"

Author Response

Response to reviewer  - 3

This manuscript is a resubmission of an earlier submission. The following is a list of the peer review reports and author responses from that submission.

Round 1

Reviewer 1 Report

This paper describes a comprehensive database of research in the area of neuroscience, especially brain diseases. The system described is based on information scraped from a number of sources, and aims to become an extensive resource for a wide range of users, scientists, and commercial interests.

Th aim is laudable and a worthwhile pursuit, but the described system is only in its infancy. This by itself is not an issue but there seems to be not enough effort put into system design and validation. The data is drawn from public accessible resources and other database collections, and stored in some relational database using MySQL, with a web interface. There is a superficial relation diagram provided, but no analysis of the data content, and data relations, and no apparent attempts to create a relational database structure reported in the manuscript. The validity of the data collection is illustrated using some examples, but there is no way to know how representative these are within the dataset nor if these are of sufficient complexity to evaluate data queries. The source of the data is also very much a concern. It is correctly reported that the issue with data collection is a lack of accessibility, annotations, data format etc, but not how the design of the system is intended to resolve any of those problems. Further issues with the database that should be addressed are the issues of data normalisation for each category, relational constraints, and the means by which data queries are handled within each subdomain.

As should be by now well understood, scraping the World Wide Web for data does not a good data collection make. Some curation of the data, verification and mechanisms for validation should be constructed. A system in which the useful data is hidden behind varying amounts of dross is no use at all.

It is also clear that attempts of this nature, to create representative data sources for a wide range of users, should make significant effort to draw on some wider community of the target audience. Without widespread support and use of the system, it will not evolve, nor grow to the intended size. Often these approaches to build a system are linked to existing organisations, such a federations of researchers within a domain.

If the authors would like to create a standardised means for data exchange and long term storage, they should also consider the actual needs of the users and aim to develop some international recognised standard for data exchange of this nature, or at least attempt to conform to existing standards (such as ISO and IEEE standards, where appropriate).

Finally, there are potential issues with the ethical constraints of such a collection, in particular, it is not explained what the data management protocols are to satisfy legal constraints, as the data is not all locally produced. There are also potential copyright issues, and the means by which the organisers intend to resolve conflicts and legal challenge of their collection. An approach used by wikipedia or open source data may well be suitable, but should be discussed and implemented appropriately.  

I hope that the authors will be able to overcome the substantial problems involved in developing such a comprehensive system, and develop this into a major new resource, which I certainly would like to explore. This manuscript is just too disjointed to be acceptable in its current form.

Author Response

Dear Editors and Reviewers,

Thank you for your email and for the reviewers’ comments concerning our manuscript entitled “Brain Research Hotspot Database (BRHD): A Panoramic Data-Base of Latest Research Hotspots for Brain” (ID: brainsci-1470914). The comments are constructive and very helpful for revising and improving our paper, as well as the important guiding significance to our researches. Each suggested revision and comment, brought forward by the reviewers was accurately incorporated and considered, which we hope meet with approval. The comments of the reviewers are response point by point and the revisions are highlighted in yellow in the manuscript.

Sincerely,

Jian Li

Southeast

Reviewer 2 Report

Title: Brain Research Hotspot Database (BRHD): A Panoramic Data-2 Base of Latest Research Hotspots for Brain

Manuscript ID: brainsci-1470914

Comments to authors

The paper introduces a database for brain science research, named Brain Research Hotspot Database (BRHD), and makes it available to the public. Authors used natural processing techniques and manual processing to integrate data from multiple data sources covering scientific papers, patents, etc. The database also offers three functions/services, namely the brain disease-genome network, brain and gut microbiome, and 3D brain structure study. In my opinion, the paper is well-written and the database could be of interest to the scientific community. However, I have some concerns/questions, as listed below, before deciding on the paper.

Comments

  • I assume BRHD is integrating data from open-access data sources. Is my assumption correct or there are also some, for example, commercial/private data sources that are integrated?
  • In the case of integrating open-access data sources, what is the advantage of BRHD over already available data sources/databases? Authors are encouraged to make detailed comparisons with similar services, elaborating more on the cons and pros of BRHD. Referring to Figure 1, more elaboration on the steps would be useful. What is unique about this interface (hotspot)?
  • BRHD allows users to search the database content. Is bulk download allowed? For example, if some articles are listed as a result of a search query, can a user download them and analyze them locally or they can be only navigated inside the web interface.
  • Some processes in BRHD are heavily manual, such as crawling, verifications, … How does this affect the scalability of the proposed database.
    • Is crawling of the target data sources permitted? For example, is it OK to crawl Web of Science?
  • Referring to Figure 5-b, is it only available in Chinese? Are there some services/pages that are only available in Chinese? If yes, offering an English page would be suggested to open up the services to the worldwide community.
  • Authors mentioned in the article that they have used 2 researchers for verification. What rule was applied in case of a tie between these 2 researchers? In the general case, it would be better to use an odd number of experts to prevent tie situations. On the same note, more elaboration on the verification process is needed. Details are encouraged.
  • A section could be dedicated to explaining the maintenance plan of BRHD in detail.
  • The database stats date back to June 2021. It would be good to update them.

Author Response

Dear Editors and Reviewers,

Thank you for your email and for the reviewers’ comments concerning our manuscript entitled “Brain Research Hotspot Database (BRHD): A Panoramic Data-Base of Latest Research Hotspots for Brain” (ID: brainsci-1470914). The comments are constructive and very helpful for revising and improving our paper, as well as the important guiding significance to our researches. Each suggested revision and comment, brought forward by the reviewers was accurately incorporated and considered, which we hope meet with approval. The comments of the reviewers are response point by point and the revisions are highlighted in yellow in the manuscript.

Sincerely,

Jian Li

Southeast University

Responds to the reviewer’s comments:

Reviewer 3 Report

This manuscript presents a panoramic database of the latest research hotspots for brain, which sounds be very useful to the brain research community. The timing of this work is excellent. I have several concerns.

(1) The database (http://117.73.3.205:8083/) is not accessible. It will be great if this database provides a batch download function.

(2) Does the database contains data from both human and model organisms? If yes, how they are organized?

(3) Page 5 lines 156-158: How annotations were done? Each entry was checked by how many annotators? If there was a disagreement between/among annotators, what procedure was taken to reconcile it.

(4) Section 2.3.4: Are the transcription datasets annotated with brain regions, cell types, and so on? 

(5) Section 3.1 lines 260-263: Will BRHD be continuously synched with other data resources? If yes, how to automate the synchronization process as much as possible?

(6) The caption of Figure 6 is confusing.

Author Response

Dear Editors and Reviewers,

Thank you for your email and for the reviewers’ comments concerning our manuscript entitled “Brain Research Hotspot Database (BRHD): A Panoramic Data-Base of Latest Research Hotspots for Brain” (ID: brainsci-1470914). The comments are constructive and very helpful for revising and improving our paper, as well as the important guiding significance to our researches. Each suggested revision and comment, brought forward by the reviewers was accurately incorporated and considered, which we hope meet with approval. The comments of the reviewers are response point by point and the revisions are highlighted in yellow in the manuscript.

Sincerely,

Jian Li

Southeast University

Round 2

Reviewer 1 Report

I am happy with the changes made and the reply to my comments addresses my concerns.

Reviewer 3 Report

This revision addresses most of my previous concerns. I tried to register an account at http://www.brainresearch.cn, however, without success. Please fix it.

Author Response

Dear Editors and Reviewers,

Thank you very much for the excellent constructive comments. We sincerely thank the reviewer for thoroughly examining our manuscript and providing very helpful comments to guide our revision. The responses to the comments are given below.

Sincerely,

Jian Li

Southeast University

Round 3

Reviewer 3 Report

Following the instruction, I was still not able to register an account from the US. The system kept giving me an error msg: "Email format does not match"